# Analysis of the Effectiveness of the Step Vacuum Preloading Method: A Case Study on High Clay Content Dredger Fill in Tianjin, China

**Jinfeng Li, Huie Chen** **, Xiaoqing Yuan \* and Wenchong Shan**

College of Construction Engineering, Jilin University, No.938 Ximinzhu Street, 130026 Changchun, China; jfli17@mails.jlu.edu.cn (J.L.); chenhe@jlu.edu.cn (H.C.); shanwc19@mails.jlu.edu.cn (W.S.)
\* Correspondence: yuanxiaoqing@jlu.edu.cn

**Abstract:** As a solution to avoid the blockage of the drainage pipe by traditional vacuum preloading, step vacuum preloading (SVP) has been progressively studied. However, the effectiveness of this technique has yet to be systematically analyzed. In this study, an indoor model test was conducted in which vacuum pressure was applied in five stages (10, 20, 40, 60, and 80 kPa) to dredger soil with high clay content at a reclamation site in Binhai New Area, Tianjin, China. The extent of the consolidation effect of the soil was determined, and the effectiveness of the step vacuum preloading method to address drainage pipe blockage was evaluated. The results indicate that soil settlement increases at each stage of vacuum pressure treatment and the degree of vertical consolidation at each stage exceeds 90%. At the end of the treatment stage with vacuum pressure of 80 kPa, the weakly bound water was discharged. Dissipation of pore water pressure occurred in all stages. On the basis of these results, it is shown that SVP can efficiently reinforce dredger fill. Moreover, after SVP, the grain size of the soil and void ratio are still uniformly distributed. Regardless of their location from the drainage pipe, soil exhibits permeability coefficients within the same order of magnitude. The consolidation effect of soil in each stage and the increased drainage rate in the initial stage of vacuum preloading with 80 kPa indicate that the test in the current study can decrease the horizontal displacement of fine particles and can avoid drainage pipe blockage.

**Keywords:** step vacuum preloading; dredger fill; permeability coefficient; consolidation effect; drain pipe blockage

## 1. Introduction

In 1952, the Swedish scholar Walter Kjellman proposed vacuum preloading. This technique has been used in many countries for various large-scale projects after decades of research and experimentation. These projects include land reclamation projects and subgrade projects in which the soil mass needs to be strengthened. The technique is low cost, requires a short period to complete construction, requires no filling materials, involves no heavy machinery, and provides good reinforcement [1–7]. Regardless, the traditional vacuum preloading (TVP) method still has several disadvantages: On the one hand, under the action of vacuum pressure and water flow, fine soil particles gradually migrate to the vicinity of the prefabricated vertical drainage pipe, forming a low-permeability layer near the drainage pipe. This process reduces the vacuum pressure on the soil, causing drainage pipe blockage and a reduction in drainage capacity [8–10]. On the other hand, owing to the temporal and spatial effects of vacuum suction transfer in space, the process can easily lead to the uneven settlement of soil mass [11]. Numerous solutions to these problems have been proposed in various studies. Adding coarse sand, lime, or $FeCl_3$ to soft soil can influence flocculation, significantly improve

the drainage capacity of the soil, shorten its consolidation time, and solidify heavy metals in soft soil. However, this method is too expensive and complex in high-scale projects and may cause secondary pollution, with even worse effects in coastal areas [12–14]. Compared with the traditional vacuum preloading technique, the combination of electroosmosis and vacuum preloading is more efficient and produces higher negative pore water pressures but entails considerably higher costs [15]. The low conductivity of the dredger fill in some cases, such as drainage to a low moisture content, renders this method unsatisfactory. Moreover, the metal electrode in the cathode is easily corroded by seawater, which also leads to the blockage of drainage pipes [16,17].

In 2006, Qing Wang (China) presented the step vacuum preloading (SVP) method. The layer-by-layer application of vacuum pressure alleviates the rapid mass migration of fine particles and fundamentally improves TVP. The advantages and disadvantages of SVP have been studied in recent years. Wang et al. [13] conducted a two-stage (at 40 and 80 kPa) vacuum preloading test [18] and a three-stage (at 20, 40, and 80 kPa) vacuum preloading test [19]. Compared with the direct vertical-vacuum preloading test, the step vacuum preloading method can improve the consolidation efficiency of the hydraulic fill and significantly reduce the blockage of the drainage pipe. Wu et al. [20] explored different approaches to vacuum loading and determined that small-vacuum loading followed by large-vacuum loading on soft soil can more easily allow the soil to settle, compared with direct large-vacuum loading; moreover, the final strength of the soil is greater. Yan [21] then explored a five-stage (10, 20, 40, 60, and 80 kPa) vacuum preloading test, and porosity testing of soil microstructure indicated a change in the soil sample from nondirectional to directional and ultimately to nondirectional. The change in directionality can indirectly reflect the drainage situation and the degree of reinforcement. The change in moisture content and drainage capacity of soil under SVP was examined by Liu et al. [22]. The results show that SVP can significantly reduce moisture content in soil, and drainage in SVP is similar to that in TVP, including the rapid growth area, slow growth area, and stable area. Using the indoor model experiment of SVP, Yuan [23] demonstrated that moisture content in soil decreased in each stage and that bearing capacity increased with a decrease in moisture content, which proved that the SVP method exerts a consolidation effect in each stage. Fang et al. [24] conducted an SVP test to apply vacuum pressure at the subsequent level on the basis of time rather than the completion of consolidation in each stage. Compared with TVP, the SVP method required longer to complete and exhibited a lower initial drainage rate. However, the accumulated drainage volume and final settlement were larger.

Changes in the soil consolidation state, moisture content, drainage, and pore water pressure during step vacuum preloading have been reported. However, the correlation between these properties has yet to be effectively analyzed. The settlement, drainage rate, pore moisture content and type, pore water pressure, and so on are interrelated and influence one another during reinforcement. Thus, a comprehensive analysis of the change law of various properties should be conducted. In addition, blockage of the drainage pipe is essentially attributable to the low permeability coefficient of soil to a certain extent, impeding water outflow. The change in permeability of soil is one of the important factors affecting the consolidation. In the study of soil treatment by vacuum preloading, the permeability of soil during TVP was evaluated. Indraratna et al. [25] conducted finite element analysis and numerical simulations by using the ABAQUS (ABAQUS, Inc., Providence, Rhode Island, USA) software and presented the conversion formula of permeability and vacuum preloading pressure under axisymmetric and equivalent plane strain conditions. However, with respect to soil permeability, the consolidation of SVP on soil with high clay content is rarely reported.

With the aforementioned problems considered, this study uses a five-stage vacuum preloading method to reinforce a coastal dredger fill with high clay content in Tianjin, China. The settlement, drainage rate, moisture content, and pore water pressure during consolidation were monitored in real-time. Moreover, the grain size distribution, void ratio, and permeability testing were conducted, and these characteristics were used to analyze the effectiveness of SVP in reinforcing the dredger fill with high clay content.

## 2. Soil Properties

The dredger fill used in this test was from the southeast bank of the Nangang Industrial Zone, Binhai New Area, Tianjin, China (E 117°37′25.84″, N 38°42′16.03″), a large-scale land reclamation site that started in 2016. Sampling was conducted in November 2017. The site was originally at the junction of a shallow sea and tidal flat and was artificially filled into land (Figure 1). Currently, the site is relatively flat in general, with slight fluctuations in several portions. The dredger fill has not been vacuum-preloaded, but the surface layer has been dried and exhibits a certain strength. Two soil samples were selected in this area for mixing; a spade was used to remove the hard surface layer from the soil. The soil taken by digging contained moisture (30–80% and up to 120%) and was transferred into a bag for sealing.

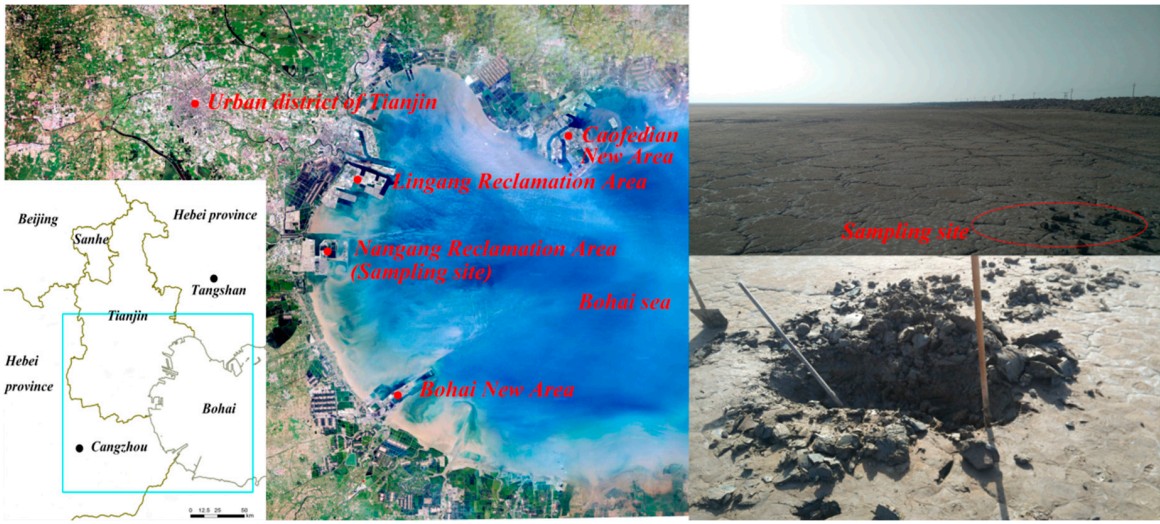

**Figure 1.** Sampling sites of the soft soil used in this study.

The basic physical properties of the dredger fill sample are listed in Table 1. Figure 2 shows the particle distribution curve of the soil sample after desalting treatment, which contained clay (52.06%) and silt (47.91%). The soil was classified as lean clay in accordance with the Standard Practice for Classification of Soils for Engineering Purposes (ASTM D2487-17). The nonuniformity coefficient $C_u = 24.17$ and the curvature coefficient $C_c = 2.06$ indicate that the soil is well-graded and that the particle size distribution is continuous. The crystal minerals in the soil samples were characterized by X-ray diffraction analysis (Table 2). The results show that the primary minerals of the soil mainly consist of quartz and calcite, the secondary minerals have high illite-smectite layer content, and the hydrophilicity is high, indicating that the water could not be easily drained from the soil.

**Table 1.** Basic physical properties of the soil sample.

| Specific Gravity | Plastic Limit (%) | Liquid Limit (%) | Soluble Salt Content (%) | pH |
|---|---|---|---|---|
| 2.74 | 26 | 45 | 1.756 | 7.12 |

**Table 2.** Mineral composition of the soil sample.

| Secondary (Clay) Mineral (%) | | | | Primary Mineral (%) | | | | | |
|---|---|---|---|---|---|---|---|---|---|
| Kaolinite | Illite | Chlorite | Illite–Smectite layer | Quartz | Plagioclase | Hornblende | Calcite | Muscovite | Alkali Feldspar |
| 3.2 | 8.0 | 4.16 | 16.64 | 36.1 | 10.3 | 0.7 | 15.8 | 2.2 | 2.9 |

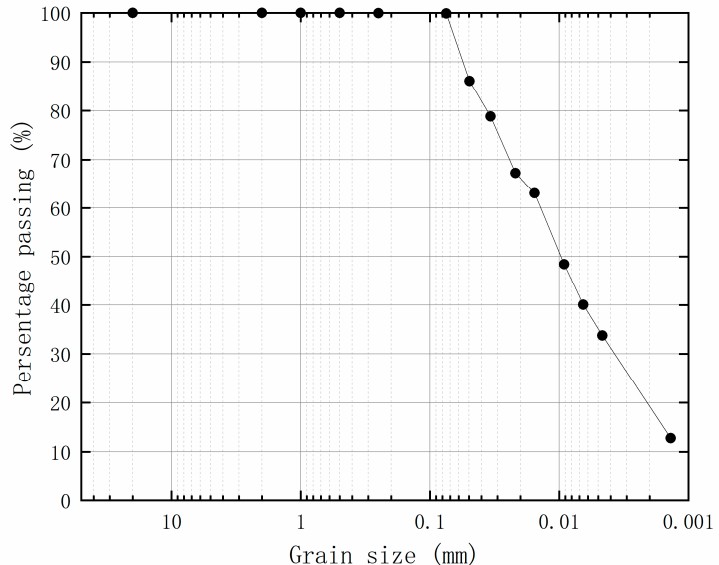

**Figure 2.** Grain size distribution of the dredger soil.

## 3. Testing Methods and Procedure

### 3.1. Consolidation Testing

As shown in Figure 3, the experimental device mainly included three parts: the main body of the settler, the drainage and air extraction system, and the observation system. The settler was a cylindrical metal bucket measuring 70 cm in diameter and 50 cm in height. The inner surface of the settler was wrapped with two layers of plastic film to isolate and seal the soil. The drainage and air extraction system included a drain board, a drain pipe, a drain barrel, and a vacuum pump. The observation system consisted of a settlement scale (observation of the settlement), a vacuum gauge (real-time detection of vacuum degree), and a pore water pressure gauge.

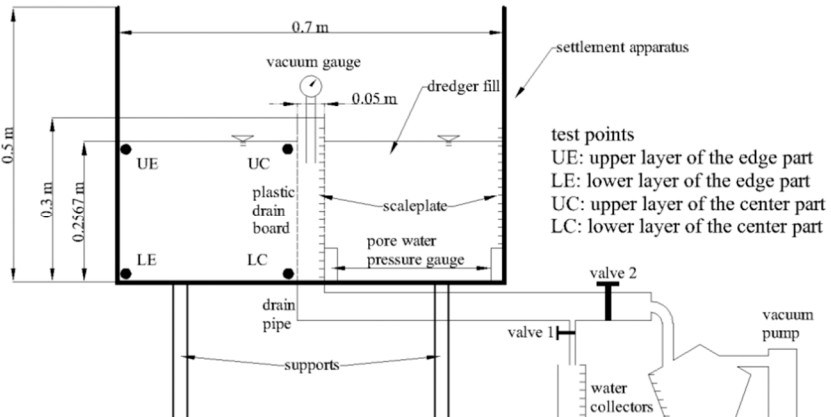

**Figure 3.** Schematic of the experimental apparatus.

In the actual consolidation process of the project, the dredger fill undergoes self-weight consolidation while drying, that is, soil-water separation and self-weight sedimentation [23]. In this indoor simulation test, the soil samples without desalting were dried, crushed, prepared, according to a moisture content of 120%, fully stirred, and soaked for 24 h. The mud was then poured into the test barrel. With this method of refilling, the settlement and change process of the dredger fill can effectively be studied [26].

Under the influence of self-weight stress, soil precipitation and water were separated. In this stage, both drain valves were closed. The soil-water separation stage ended when the soil and water surfaces became stable. Drain valve 1 was then opened for the self-weight sedimentation stage. The water in the settler was discharged along the drainage pipe, and the drainage volume was determined. The self-weight sedimentation stage ended when the soil surface coincided with the water surface, the height of the mud surface did not change within 48 h, and the pore water pressure stabilized at a certain value. The soil-water separation and self-weight sedimentation stages are referred to as the self-weight consolidation stage.

After the self-weight consolidation stage, SVP testing was started. In this stage, the plastic film on the inner surface of the settler separated the soil from the outside; thus, only the drainage pipe could discharge the water in the settler. Drain valve 1 was closed, drain valve 2 was opened, and the water discharged from the water collector was calculated. The vacuum pump was used to apply vacuum pressure to the whole system. Five levels (10, 20, 40, 60, and 80 kPa) of vacuum pressure were applied based on the testing time sequence. The consolidation stability standard of the dredger fill under the action of vacuum suction in each stage was a soil surface settlement of less than 1 mm within 48 h, meaning that the consolidation of soil in this stage of pressure was completed. Even if increasing consolidation time had no further effect on consolidation [27], the next level of vacuum pressure could be applied.

### 3.2. Monitoring Testing

During testing, the settlement, drainage rate, and pore water pressure of the soil in the test cylinder were monitored. At the end of each stage of vacuum preloading, the moisture content and permeability of the soil samples were determined. At the end of SVP, the grain size distribution and void ratio at different positions of the soil were tested. The results of the sampling and test scheme are listed in Table 3. The inclinometer was embedded at the edge of the settler to measure the horizontal displacement. The horizontal displacement of the upper surface at the edge was considered as the total maximum horizontal displacement. Owing to the high moisture content and poor consolidation effect of the soil in the vacuum phase of SVP at 10 kPa, sampling and permeability testing were not conducted in the vacuum phase of the technique at 10 kPa. The soil at the bottom of the settler was also neither sampled nor evaluated to prevent disturbance. The TST-55 model penetrameter (Nanjing Soil Instrument Factory Co., Ltd., Jiangsu Province, China) was used for penetration testing (Figure 4). At the end of each stage, samples were taken with a 61.8 mm diameter and 40 mm high ring knife, in accordance with the results listed in the sampling table. The ring cutter with a penetration sample was put into the saturator for saturation and then placed in the permeameter to adjust the water head for testing.

**Table 3.** Soil sampling and testing schemes.

| Test stage | Post-20 kPa | | Post-40 kPa | | Post-60 kPa | | Post-80 kPa | |
|---|---|---|---|---|---|---|---|---|
| Time (d) | 68.13 | | 83.69 | | 99.98 | | 116.37 | |
| Test Types | Moisture Content | Permeability | Moisture Content | Permeability | Moisture Content | Permeability | Moisture Content | Permeability |
| UC | √ | √ | √ | √ | √ | √ | √ | √ |
| LC | √ | – | √ | – | √ | – | √ | √ |
| UE | √ | √ | √ | √ | √ | √ | √ | √ |
| LE | √ | – | √ | – | √ | – | √ | √ |

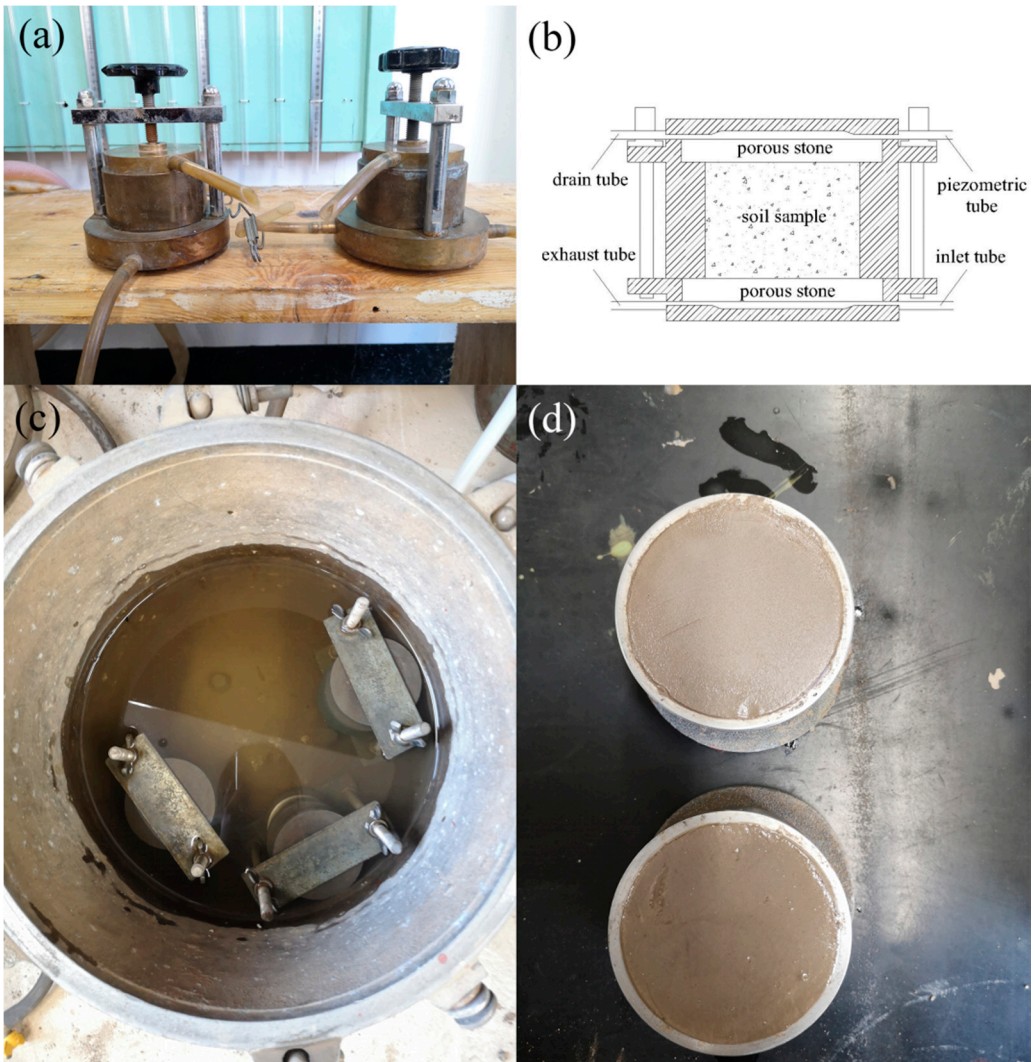

**Figure 4.** (**a**,**b**) TST-55 permeameter, (**c**) soil samples saturated in a saturator, and (**d**) soil samples after permeability testing.

## 4. Test Results and Discussion

### 4.1. Settlement

Figure 5 shows the process of settlement in each stage and Table 4 shows the specific data of settlement in each stage. When mud was poured into the settler, the initial height of the soil surface from the bottom plate was 25.67 cm. The total water volume did not change during soil-water separation, and the water surface settlement was 1.39 cm. This small settlement resulted from the flattening of the plastic film by the mud and the adjustment of the mud position, which rendered the mud fully filled in the settler. However, at the beginning of this stage, the soil surface decreased rapidly and gradually stabilized owing to the accumulation of soil particles and filling between particles. The soil itself exhibited uniform particle size distribution, resulting in good gradation and satisfactory filling between soil particles. The soil surface sank considerably, reaching 8.04 cm. After soil-water separation, self-weight sedimentation occurred. In this stage, the water surface decreased rapidly and overlapped again with the soil surface after a certain time, owing to the opening of the drain valve. Consequently, the water surface was no longer higher than the soil surface; thus, the change in water surface was no longer recorded. When the soil surface and water surface coincided, the settlement and time showed a linear relationship, and soil drainage continued. The settlement of the soil surface was

1.33 cm during self-weight sedimentation, and all occurred after the soil surface coincided with the water surface. This occurrence indicated that the water above the soil surface was discharged before the coincidence, and the water between the soil pores was discharged after the coincidence, resulting in soil settlement.

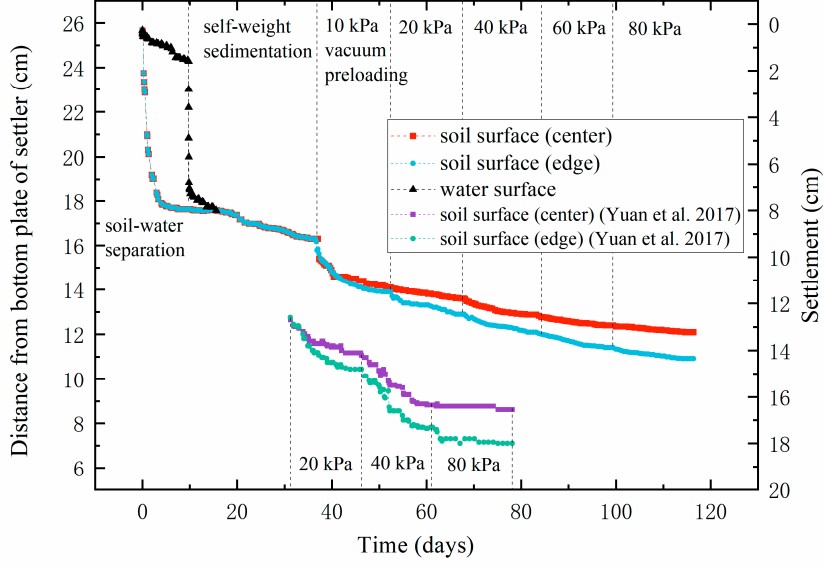

**Figure 5.** Time-dependent settlement curves in different stages.

**Table 4.** Soil sampling and testing schemes.

| Various Stages | End Time (d) | Observation Position | Water/Soil Surface Height (cm) | Accumulated Settlement (cm) | Settlement in Each Stage (cm) |
|---|---|---|---|---|---|
| Soil-water separation | 9.74 | Water surface | 24.28 | 1.39 | 1.39 |
|  |  | Soil surface | 17.63 | 8.04 | 8.04 |
| Self-weight sedimentation | 36.56 | Water surface | 16.3 | 9.37 | 7.98 |
|  |  | Soil surface | 16.3 | 9.37 | 1.33 |
| 10 kPa | 52.43 | Soil surface (center) | 14.1 | 11.57 | 2.2 |
|  |  | Soil surface (edge) | 13.9 | 11.77 | 2.4 |
| 20 kPa | 68.13 | Soil surface (center) | 13.6 | 12.07 | 0.5 |
|  |  | Soil surface (edge) | 12.9 | 12.77 | 1 |
| 40 kPa | 83.69 | Soil surface (center) | 12.88 | 12.79 | 0.72 |
|  |  | Soil surface (edge) | 12.1 | 13.57 | 0.8 |
| 60 kPa | 99.98 | Soil surface (center) | 12.4 | 13.27 | 0.48 |
|  |  | Soil surface (edge) | 11.4 | 14.27 | 0.7 |
| 80 kPa | 116.37 | Soil surface (center) | 12.1 | 13.57 | 0.3 |
|  |  | Soil surface (edge) | 10.9 | 14.77 | 0.5 |

During vacuum preloading, the smaller settlement of the soil surface at the center and the larger settlement of the soil surface at the edge were observed. The differential settlement increased with an increase in vacuum pressure. Moreover, the total settlement at the center was 4.20 cm, the total settlement at the edge was 5.40 cm, and the maximum differential settlement was 1.20 cm. Figure 5 presents the settlement curve of the three-stage (at 20, 40, and 80 kPa) vacuum preloading conducted by Yuan et al. [23]. The soil used for testing in the current study exhibited similar basic properties as

those reported by Yuan et al. [23], which was attributed to the similar locations of both types of soil. The vacuum-preloaded soil exhibited a consistent trend in settlement under different loading methods. The two loading techniques also effectively consolidated the soil. Compared with the test reported by Yuan et al. [23], the test in the current study required a longer settlement time but exhibited a smoother settlement curve. Moreover, the maximum differential settlement was smaller between the center and the edge than that obtained by Yuan et al. A maximum differential settlement between the center and the edge obtained by Yuan et al. [23] was 1.45 cm and a maximum differential settlement to horizontal settlement distance ratio was 0.0483. Compared with this result, the ratio determined in the present study was smaller—that is, 0.04.

Moreover, under vacuum pressure, the displacement of the soil toward the central drainage pipe resulted in a higher soil surface at the center than the edge. Horizontal displacement occurred in each stage of vacuum pressure treatment. The higher the vacuum pressure, the larger the differential settlement and the greater the cumulative horizontal displacement. By applying vacuum pressure in stages, soil displacement was entirely distributed in sections, and the uneven settlement of soil was delayed. Mesri [28] performed traditional vacuum loading on various soft clay and muddy sediments; the horizontal displacement of the surface soil and the surface settlement ratio at the center ranged from 0.2 to 0.5, with an average of 0.36. In this test, the cumulative maximum horizontal displacement and the cumulative surface settlement ratio at the center were 0.09, 0.26, 0.22, 0.25, and 0.29, respectively. The average ratio was 0.22 and the maximum ratio was 0.29. The average and maximum ratios were smaller than the values obtained in the study by Mesri [28], confirming that the loading method of this test can reduce horizontal displacement.

*4.2. Consolidation Degree*

In this study, the logarithmic curve method was used to estimate the final settlement in each stage and calculate the degree of consolidation [29]. The theoretical general solution of the degree of consolidation is as follows:

$$U = 1 - \alpha \cdot e^{-\beta t}, \tag{1}$$

where $U$ is the degree of consolidation, $e$ is the natural constant, and $\alpha$ and $\beta$ are parameters under different drainage paths and different consolidation conditions. Whether vertical drainage, horizontal outward drainage, or horizontal central drainage can be used, only the values of $\alpha$ and $\beta$ vary. The degree of consolidation is defined as

$$U = \frac{S_t}{S_\infty}, \tag{2}$$

where $S_t$ is the settlement at time $t$ and $S_\infty$ is the final settlement. $(t_1, S_1)$, $(t_2, S_2)$, and $(t_3, S_3)$ are any three points satisfying $t_3 - t_2 = t_2 - t_1$ in the measured settlement curve. These three points are substituted into Equation (1) and Equation (2), respectively, to derive the following equations:

$$\begin{cases} S_1 = S_\infty(1 - \alpha \cdot e^{-\beta t_1}) \\ S_2 = S_\infty(1 - \alpha \cdot e^{-\beta t_2}) \\ S_3 = S_\infty(1 - \alpha \cdot e^{-\beta t_3}) \end{cases} \tag{3}$$

Further, the following is obtained:

$$S_\infty = \frac{S_2^2 - S_1 S_3}{2S_2 - S_1 - S_3} = \frac{S_3(S_2 - S_1) - S_2(S_3 - S_2)}{(S_2 - S_1) - (S_3 - S_2)} \tag{4}$$

The calculated final settlement in each stage is substituted into Equation (2), and the degree of consolidation in each stage is listed in Table 5. The degree of consolidation in each stage exceeded 90%, which verifies the effectiveness of SVP.

**Table 5.** Settlement and consolidation degree in different stages.

|  | Self-Weight Consolidation | 10 kPa | 20 kPa | 40 kPa | 60 kPa | 80 kPa |
|---|---|---|---|---|---|---|
| Settlement at the center (cm) | 1.33 | 2.20 | 0.50 | 0.72 | 0.48 | 0.30 |
| Consolidation degree at the center | 98.50% | 95.65% | 92.60% | 98.23% | 98.63% | 96.77% |
| Settlement at the edge (cm) | 1.33 | 2.40 | 1.00 | 0.80 | 0.70 | 0.50 |
| Consolidation degree at the edge | 98.50% | 99.45% | 90.90% | 93.46% | 91.58% | 98.04% |

## 4.3. Drainage Rate

The drain valve was opened at the end of the soil–water separation stage—that is, at the beginning of the self-weight sedimentation stage; thus, the drainage rate was measured from this time. The drainage rates in different stages are shown in Figure 6. In the initial stage of self-weight sedimentation and vacuum preloading, the drainage rate immediately reached its maximum and then decreased. The peak value of the drainage rate during self-weight sedimentation was considerably greater than that during vacuum preloading. The reason is that the soil was deposited during soil-water separation and the distance between the soil surface and the water surface was 6.28 cm, allowing direct contact between the water above the soil surface and the drainage pipe. After the drainage valve was opened, the water was discharged rapidly, resulting from the gravitational potential energy. With a decrease in distance between the soil surface and the water surface, the drainage rate decreased to a relatively stable area and continued to decrease slowly. In the initial stage of vacuum pressure treatment, the water in the soil was forced to discharge under an external force, with a large flow rate and a large drainage rate. In each stage of vacuum pressure treatment, the bonding force between the soil and the water was gradually balanced with vacuum pressure over time, thereby reducing the drainage and drainage rate [30]. At the end of each stage, the drainage rate approached 0. This finding implied that the bonding force between the soil and water was nearly balanced with the vacuum pressure in the entire settler and that soil drainage was impeded. At the initial stage of vacuum pressure treatment with vacuum pressure of 80 kPa, the peak value of the drainage rate remained high, demonstrating that there occurred no blockage of drainage pipes.

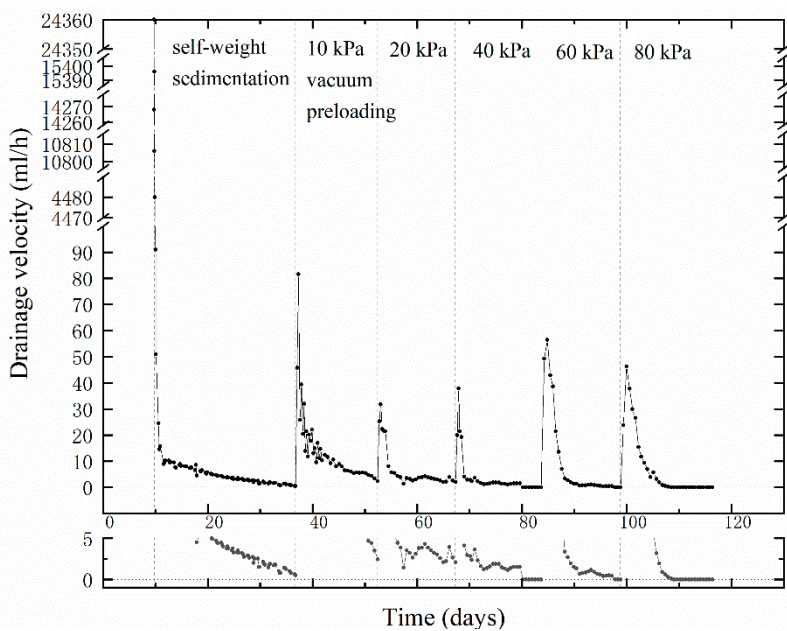

**Figure 6.** Time-dependent curve of drainage rate in the different stages of treatment.

*4.4. Moisture Content*

At the end of each stage, the change in moisture content in soil at different positions (UE, UC, LE, and LC, these locations are shown in Figure 3) was evaluated (Figure 7). During consolidation, free water was discharged first, followed by bound water. To a certain extent, the total bound moisture content adsorbed by the soil mainly consisting of clay could be characterized by its liquid limit, and the maximum moisture content of bound water in clay was close to the liquid limit [31]. In addition, the plastic limit resulted from either cavitation or air entry, preventing the water phase from acting as a continuum within the soil thread [32]. During consolidation, the moisture content in soil spanned the liquid limit and the plastic limit, which could be regarded as the drainage of soil mass from free water to loosely bound water. As shown in Figure 7, moisture content decreased sharply and then gradually. The rate of change in moisture content was particularly evident on the upper layer. In the treatment stages with a vacuum pressure of 10 and 20 kPa, the moisture content approached the liquid limit, which could be regarded as the transformation of the drainage form from free water to bound water. The moisture content on the upper layer was always lower than that on the lower layer, indicating that the transformation of free water discharged from the upper layer to loosely bound water occurred before that from the lower layer. Similarly, the conversion of free water discharged from the center to loosely bound water occurred before that from the edge. Near the end of the treatment stage with a vacuum pressure of 80 kPa, the soil moisture content in the settler approached the plastic limit and the degree of drainage was maximized without gas inflow. This finding also indicated that the loosely bound water was basically discharged, confirming that the drainage capacity of SVP was highly effective.

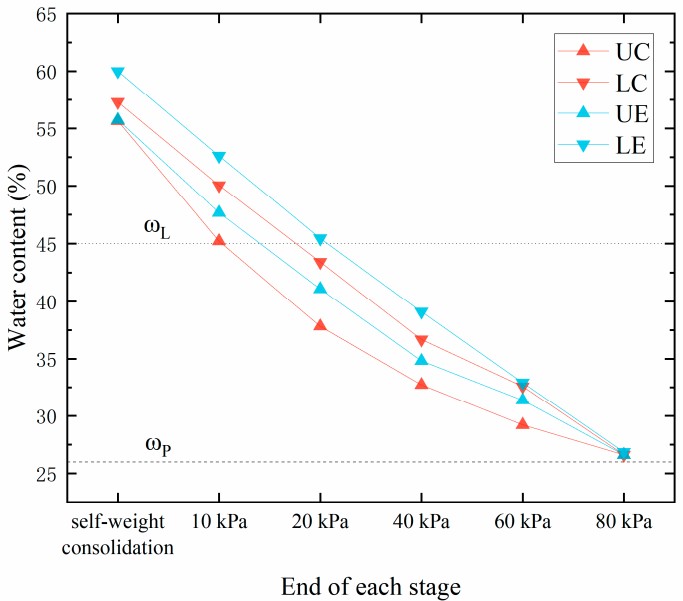

**Figure 7.** Moisture content in different locations at the end of each stage.

During self-weight sedimentation, no vacuum pressure was applied, and the water was discharged by water seepage itself. After the soil surface coincided with the water surface, the free water on the upper layer no longer exhibited the trend of transverse flow because of the absence of an upper pressure effect, and the free water among particles tended to flow vertically under the effect of gravitational potential energy. Thus, the soil moisture content at the center and the edge showed no difference, and both were smaller than that at the bottom. For the moisture content of the bottom layer, the water at the center easily flowed laterally from the drainage pipe because of the effect of gravitational potential energy of the soil and water above and the short seepage path close to the drainage pipe at the center. Thus, the moisture content at the center was slightly less than the that at the edges. In

all stages of vacuum pressure treatment, the moisture content formed a constant gradient difference, and the moisture content gradient was consistently $\omega_{UC} < \omega_{UE} < \omega_{LC} < \omega_{LE}$. During vacuum pressure consolidation, water flowed downward and toward the center under the action of gravitational potential and vacuum pressure potential. The equipotential surface in the settler was funnel-shaped, which was low at the center and high at the edge. After the consolidation settlement, the soil moisture content in the settler was equal.

### 4.5. Pore Water Pressure

Pore water pressure gauges were buried at the bottom of the center and at the edge of the settlement bucket to measure the pore water pressure in different stages in real-time. Under the influence of temperature and air pressure, the measured pore water pressure fluctuated slightly but did not affect the overall result. The dissipation of pore water pressure when vacuum pressure was applied during testing in the current study and in the study by Yuan et al. [23] is illustrated in Figure 8. Both tests exhibited similar trends in pore water pressure when the vacuum pressure changed. As shown in Figure 8, the pore water pressure was consistently lower at the center than at the edge, suggesting that when the distance of the soil to the drainage pipe was smaller, a better consolidation effect was observed. The dissipation rate of pore water pressure was lower in the current study than that in the study by Yuan et al. [23]. Similarly, the differential dissipation rate of pore water pressure between the center and the edge after the test were also lower.

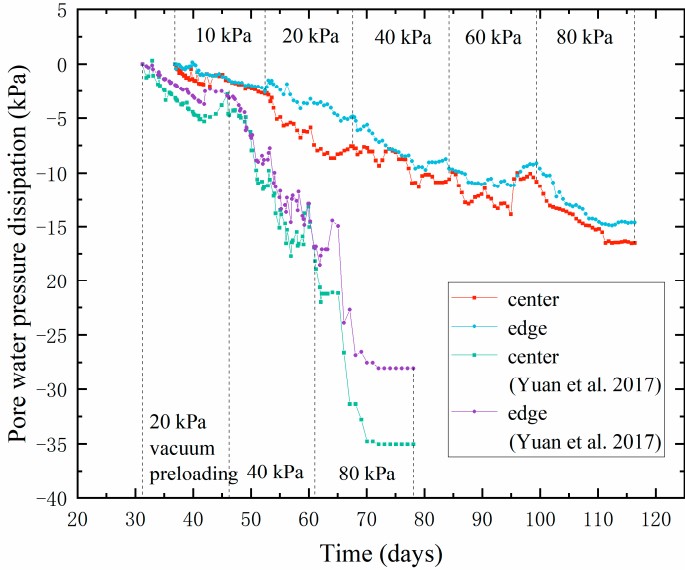

**Figure 8.** Dissipation of pore water pressure by step vacuum preloading (SVP) with different loading modes.

The change process of pore water pressure in this test is shown in Figure 9. In the initial stage of soil–water separation, the soil particles flocculated and sank, and water moved upward relatively. The drain valve was not opened, and the total water volume in the test barrel remained constant. However, when the soil-water separation particles flocculated and sank, the pores in the soil increased relatively, and the effective seepage areas largely varied when water seepage occurred. Therefore, in the initial stage of soil–water separation, the permeability per unit area was higher than the pore water pressure, resulting in an increase in pore water pressure at the beginning of the test. This effect was reflected in the pore water pressure in the initial stage of soil–water separation, which was nearly double that of its value before the process. The clear soil surface then appeared. The soil and water continued to separate, and the pore water pressure decreased slightly. During self-weight sedimentation, the water above the soil surface was discharged outward and the total water volume in the settler began

to decrease before the coincidence of soil surface and water surface. Thus, the gravity above the soil at the bottom of the settler was reduced and the drainage rate was larger relatively, resulting in an obvious decrease in pore water pressure. After this coincidence, the free water in the soil began to drain and the moisture content gradually decreased. However, the drainage rate was small, so the decrease in pore water pressure was small.

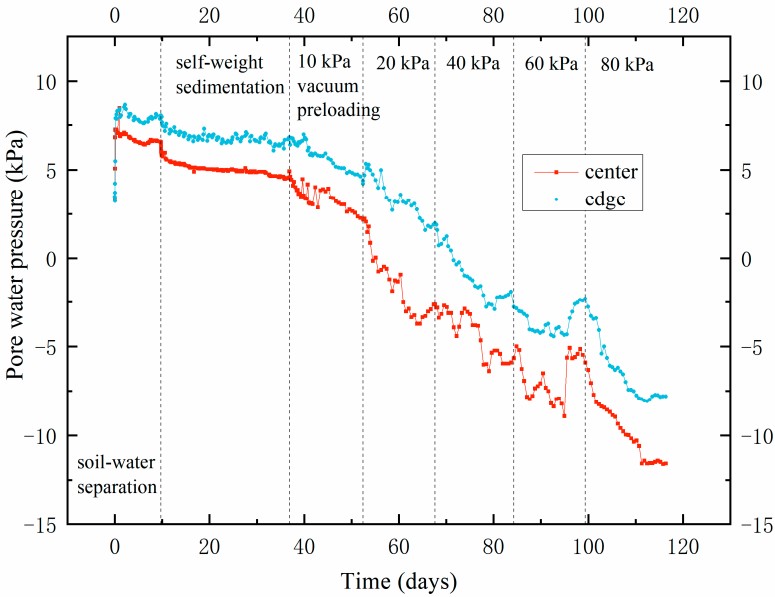

**Figure 9.** Time-dependent pore water pressure curves in the different stages of treatment.

During vacuum preloading, the pore water pressure generally exhibited a downward trend. The change in pore water pressure was not only related to the distance between drainage pipes but also to the type of water discharged from the soil [33]. As shown in Figure 8, the pore water pressure decreased steadily during the entire process when the center and the edge were in the self-weight consolidation stage and in the treatment stage with a vacuum pressure of 10 kPa. However, in the center of the final vacuum pressure stage, that is, when pressure levels of 20, 40, and 60 kPa were applied while at the edge of the final vacuum pressure stage, that is, when pressure levels of 40 and 60 kPa were applied, the pore water pressure increased slightly, which was similarly observed during the three-stage vacuum preloading test conducted by Yuan and Lei [9].

Section 4.4 of this paper indicates that the period of transition from free water at the center to loosely bound water occurred earlier than the transition at the edge. With this observation, the current study concludes that an increase in pore water pressure was a distinct phenomenon occurring during discharge of loosely bound water. By backward reasoning, when the center of the settler was in the treatment stage with a vacuum pressure of 20 kPa, transformation into loosely bound water for discharge occurred, while the edge was in the treatment stage a with vacuum pressure of 40 kPa. In Figure 7, in the later stages of vacuum pressure treatment (40, 60, and 80 kPa), the drainage rate reached 0 in advance—that is, when the whole soil mass in the settler was in the period of loosely bound water discharging outward, the soil mass in the later stage could not continue to drain outward. Combining Figures 5 and 7, when the vacuum pressure was 40 and 60 kPa, from the time when the drainage rate reached 0 until the end of each stage, the soil continued to settle by 2.2 and 2.5 mm, respectively. When the drainage rate reached 0 in the treatment stage with a vacuum pressure of 80 kPa, the settlement of the soil was stable, and the settlement curve of the soil no longer changed. Therefore, in the treatment stages with a vacuum pressure of 40 and 60 kPa, drainage stability was observed earlier in the settler, compared with the settlement stability of the soil mass. In this type of soil, which was no longer drained but still shows a small amount of settlement, the soil pore was correspondingly reduced, which caused the soil pore water pressure to slightly rise.

On the other hand, with the gradual decrease in the loosely bound water in the soil, the vacuum pressure could not further drain the water at a certain time. Owing to pore reduction, the bonding force between soil and water was slightly higher than the vacuum pressure at this time. The loosely bound water partly back flowed to create a balance between the vacuum suction and water absorption capacity of the soil; this process caused the pore water pressure in the soil to rise. Consequently, part of the loosely bound water flowed back to balance the two forces, prompting an increase in pore water pressure in soil from another angle. At the end of the treatment stage with a vacuum pressure of 80 kPa, the loosely bound water was drained, and no return flow occurred, resulting in no increase in pore water pressure.

In the current study, the decrease in pore water pressure was low in the self-weight consolidation stage and the treatment stage, with a vacuum pressure of 10 kPa, but was large in the treatment stage with a vacuum pressure of 20 and 80 kPa.

### 4.6. Grain Size Distribution and Void Ratio

Determination of the grain size distribution and void ratio of the soil was conducted after SVP. The results are presented in Table 6. Desalting was not conducted during the grain size distribution test to measure the actual grain size distribution of soil in various positions. Subsequently, the clay contents of the soil at the center and at the edge after the test were measured. A higher clay content and a lower silt content were found in the soil at the center than at the edge. Moreover, the soil at the center was more compact than that at the edge, as indicated by the void ratios. These differences were attributed to vacuum pressure. However, soil in various locations generally showed highly similar grain size distribution and void ratios, suggesting that after treatment with SVP, the soil mass was relatively even.

**Table 6.** Grain size distribution and void ratio in different positions of the soil at the end of SVP.

|  | UC | | | UE | | | LC | | | LE | | |
|---|---|---|---|---|---|---|---|---|---|---|---|---|
| **Grain Size Distribution (%)** | **Clay** | **Silt** | **Sand** | **Clay** | **Silt** | **Sand** | **Clay** | **Silt** | **Sand** | **Clay** | **Silt** | **Sand** |
|  | 33.07 | 66.05 | 0.88 | 32.97 | 66.49 | 0.54 | 34.46 | 64.11 | 1.43 | 33.39 | 65.99 | 0.62 |
| Void ratio | 0.54 | | | 0.55 | | | 0.54 | | | 0.57 | | |

### 4.7. Permeability Coefficient

Figure 10 presents the change in soil permeability in different stages of vacuum pressure treatment in the settler. In the logarithmic diagram, permeability in each stage of SVP was nearly linear. The change in the permeability coefficient appeared to decrease by an order of magnitude for each vacuum pressure stage (Figure 10). As vacuum preloading progressed, the permeability coefficient at the center was consistently smaller than that at the edge, and the difference between the permeability coefficient at the center and the edge increased. The difference in the permeability coefficient was attributable to the horizontal displacement of the soil mass toward the central drainage pipe during reinforcement, which rendered the soil sample structure at the center more dense than that at the edge. Meanwhile, under the action of vacuum pressure, fine particles in the soil migrated to the center of the drainage pipe, affecting the permeability of the soil. As permeability decreased, the permeability coefficients of the soil at the center and the edge were always close. At the end of the treatment stage, with a vacuum pressure of 80 kPa, the difference in permeability between the center and the edge was the largest but did not exceed an order of magnitude. The decrease in permeability was attributable to the reduction in porosity. The close permeability coefficient between the center and the edge indicates that the drainage capacity of the center was close to the drainage capacity of the edge.

The combined results on the similarity of the grain size distribution and void ratios of soil in various locations reveal that no blockage was observed near the drainage pipe at the center regardless of the displacement of fine particles. This observation confirms that SVP can effectively prevent drainage pipe blockage.

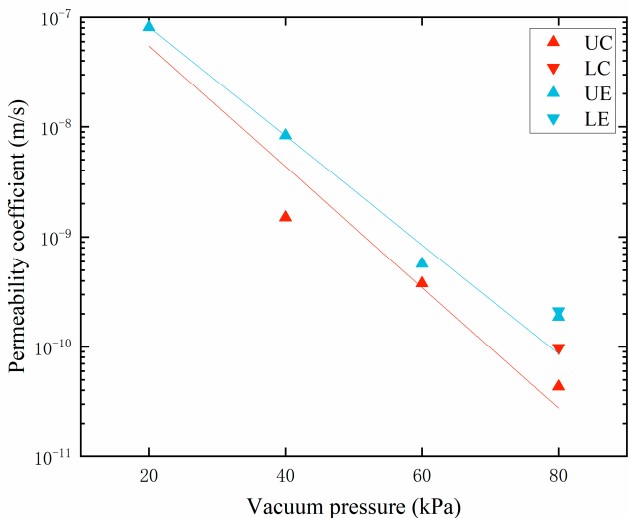

**Figure 10.** Permeability coefficient in each stage.

## 5. Conclusions

On the basis of previous research on step vacuum preloading (SVP), this study presents an experimental study of the SVP (10, 20, 40, 60, and 80 kPa) process in a coastal dredger fill in Tianjin, China. This study evaluated the consolidation effect of this test on a dredger fill with high clay content and analyzed the effective prevention of drain pipe blockage. The loading method was found to effectively consolidate and drain dredger fill with high clay content. The following conclusions are drawn based on this case study:

1.  An increase in the vacuum pressure stage can lengthen the process of settlement but can also reduce the differential settlement. In each stage of vacuum preloading, the degree of consolidation can exceed 90%. With SVP, the horizontal displacement of soil can be reduced. As vacuum pressure stages increase, the volume of horizontal displacement decreases.
2.  Owing to the different forces and drainage paths of pore water at different positions, the moisture content in soil near the drain pipe is always less than that in soil far away from the drain pipe in the vacuum preloading stage; moreover, moisture content in soil on the upper layer is less than that on the bottom layer. The relationship of moisture content, Atterberg limit, drainage rate, and pore water pressure indicates that free water is discharged from the soil mass in the treatment stage, with a vacuum pressure of 10 kPa. Loosely bound water is discharged from the center in the 20 kPa vacuum pressure stage. The soil mass of the settler is in the stage of loosely bound water discharge, and all loosely bound water is discharged at the end of the treatment stage with a vacuum pressure of 80 kPa.
3.  With an increase in the vacuum pressure stage, the dissipation rate of pore water pressure tends to be even, regardless of the position. The pore water pressure decreases during the entire process, and the dissipation at the center is greater than that at the edge. At the later stage of some vacuum pressure treatment, the pore water pressure slightly rises, but no discharge of loosely bound water occurs. This phenomenon can be attributed to the compaction of the soil and the backflow of loosely bound water.
4.  SVP is an effective means of avoiding drainage pipe blockage, as verified by the consolidation effect of the soil in each stage, as well as by the increase in drainage rate in the initial stage of vacuum preloading with 80 kPa. Similarities between the center and the edge, with respect to the grain size distribution, void ratio, and permeability coefficient of the soil, indicate that the distance from the drainage pipe causes no horizontal variation in fine particles and that the consolidation effect exerted is even.

**Author Contributions:** Conceptualization, X.Y.; methodology, W.S.; software, J.L.; formal analysis, J.L.; resources, H.C.; data curation, W.S.; writing—original draft preparation, J.L.; writing—review and editing, J.L.; funding acquisition, X.Y. and H.C. All authors have read and agreed to the published version of the manuscript.

**Funding:** The presented work is supported by the National Natural Science Foundation of China (No. 41602285, No. 51890914), Science and Technology Development Program of Jilin Province, China (No. 20180520064JH).

**Acknowledgments:** The authors would like to thank the anonymous reviewers for their constructive comments and suggestions.

**Conflicts of Interest:** The authors declare no conflict of interest.

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
