# Peer review of "Analysis of the Effectiveness of the Step Vacuum Preloading Method: A Case Study on High Clay Content Dredger Fill in Tianjin, China"

_jmse, doi:10.3390/jmse8010038_

Round 1
Reviewer 1 Report
End of line 61, I think "expore" should be "explore"
Line 66, "The change in moisture content and drainage capacity of soil under SVP was examined [22]", should be changed to "examined by ..... [22]".
Line 82, remove the word "effect" at the end of the sentence, so sentence should end by "consolidation."
There is a leap/disconnection of concept between line 86 and 87, perhaps has to do with sudden introduction of "reinforcement effect of SVP"!
Must remove line 91, where it states the "Change rules and correlation of each parameter in different stages are analyzed". Paper can not make such a claim. It does not describe what change rules and what form of correlation, or that in either case what was the need (lack of information), what is your hypothesis (expectation), was the analysis formed to validate existing rules and correlations or new forms (models) with accompanying parameters are formulated as a result of this study (analysis). Since neither is done, the statement gives a false sense as if study validates or formulates these, and it is misleading.
Clearly state that vacuum consolidation samples are not prepared from the desalted soil samples.
Line 336, chapter 4.4 of what?
Author Response
Comment 1:End of line 61, I think "expore" should be "explore"
Response:Thank you for your suggestions. The word "expored" in line 61 has been changed to "explored"
Comment 2:Line 66, "The change in moisture content and drainage capacity of soil under SVP was examined [22]", should be changed to "examined by ..... [22]".
Response:Thank you for your suggestions. The sentence "The change in moisture content and drainage capacity of soil under SVP was examined [22]" has been changed to "The change in moisture content and drainage capacity of soil under SVP was examined by Liu et al. [22]."
Comment 4:Line 82, remove the word "effect" at the end of the sentence, so sentence should end by "consolidation."
Response:Thank you for your suggestions. The word "effect" has been deleted and the sentence has been changed to "The change in permeability of soil is one of the important factors affecting the consolidation."
Comment 5:There is a leap/disconnection of concept between line 86 and 87, perhaps has to do with sudden introduction of "reinforcement effect of SVP"!
Response:Thank you for your suggestions. This sentence corresponds to "The change in permeability of soil is one of the important factors affecting the consolidation" between line 83 and 84. So "reinforcement effect of SVP" has been changed to "consolidation effect of SVP".
Comment 6:Must remove line 91, where it states the "Change rules and correlation of each parameter in different stages are analyzed". Paper can not make such a claim. It does not describe what change rules and what form of correlation, or that in either case what was the need (lack of information), what is your hypothesis (expectation), was the analysis formed to validate existing rules and correlations or new forms (models) with accompanying parameters are formulated as a result of this study (analysis). Since neither is done, the statement gives a false sense as if study validates or formulates these, and it is misleading.
Response:Thank you for your suggestions. The sentence "and the change rules and correlation of each parameter in different stages were analyzed" has been removed.
Comment 7:Clearly state that vacuum consolidation samples are not prepared from the desalted soil samples.
Response:Thank you for your suggestions. It has been mentioned in line 132 and the sentence "In this indoor simulation test, the soil samples are dried, crushed, and prepared according to a moisture content of 120%, fully stirred, and soaked for 24 h" has been changed to "In this indoor simulation test, the soil samples without desalting are dried, crushed, and prepared according to a moisture content of 120%, fully stirred, and soaked for 24 h."
Comment 7:Line 336, chapter 4.4 of what?
Response:Thank you for your suggestions. This sentence is intended to refer to the conclusion drawn in lines 277 to 288 of chapter 4.4 of this paper. So the sentence "Chapter 4.4 indicates that the period of transition from free water at the center to loosely bound water occurs earlier than the transition at the edge" has been changed to "Chapter 4.4 of this paper indicates that the period of transition from free water at the center to loosely bound water occurs earlier than the transition at the edge."

Reviewer 2 Report
The revised version has been improved significantly. I have no more comments. The article can be accepted for publication without further modification.
Author Response
Comments of Reviewer 2: The revised version has been improved significantly. I have no more comments. The article can be accepted for publication without further modification.
Response:Thank you for your recognition of our work and we will work harder in the future.

Reviewer 3 Report
Dear Authors,
This is the 2nd time that I review this article. The case-study nature of the work is now better communicated in the title. However, 90% of my previous comments in the PDF file have been ignored. Why can this obvious Grammar issues and inconsistencies not be removed? Do the Author not have access to the PDF file (use Adobe Acrobat Reader)?
This is the last chance for the Authors to make an effort to improve their work based on the comments in the attached PDF file. Otherwise, I will have to recommend that the work is rejected.
Kind regards,
Reviewer

Author Response
Comments of Reviewer 3:
This is the 2nd time that I review this article. The case-study nature of the work is now better communicated in the title. However, 90% of my previous comments in the PDF file have been ignored. Why can this obvious Grammar issues and inconsistencies not be removed? Do the Author not have access to the PDF file (use Adobe Acrobat Reader)?
This is the last chance for the Authors to make an effort to improve their work based on the comments in the attached PDF file. Otherwise, I will have to recommend that the work is rejected.
Response:Thank you for your suggestions. We are awfully sorry that your previous comments were ignored, because this PDF file has not been found on the platform before. We tried our best to improve the manuscript based on your comments and made some changes in the manuscript. And here we did not list the changes but marked in red in revised paper. We appreciate for your warm work earnestly, and hope that the correction will meet with approval. Once again, thank you very much for your comments and suggestions.

This manuscript is a resubmission of an earlier submission. The following is a list of the peer review reports and author responses from that submission.
Round 1
Reviewer 1 Report
The study claims improved response based on 5 stage implementation of SVP.
The discussion and results appear to be based on a single test. Sampling effects (carried out at the end of the 4 stages), such as the impact of interruption (if any) of the vacuum process on various measurements may effect the explanations and discussion of observations of pwp and other measurements.
Recommendations:
Conduct the same study (SVP) more than once, to verify measured pwp and settlement responses. Utilize several PWP spaced radially from the center. Avoid sampling during these SVP to justify explanations on pwp response Conduct a single stage High Vacuum test on a sample that is prepared in a similar manner to provide baseline and validate the effectiveness and improvements stated in the study.
Author Response
Thank you very much for your valuable and insightful suggestions. Conducting the same study (SVP) more than once to verify measured PWP and settlement responses adds rigor to the study. However, one SVP test takes several months to complete. Because of the limited experimental conditions, conducting multiple tests in this study would be impractical. In the preparation of the experiment, our team conducted a series of feasibility analyses of the experimental scheme. In 2004, Indraratna et al. discussed the effect of vacuum removal and reapplication during vacuum preloading and found that the interruption of vacuum pressure exerted no effect on the results of settlement and other data. Yuan [23] and Fang [24] indicated that sampling and testing during step-vacuum preloading can help determine the corresponding change laws on water content, pore water pressure, and so on. Moreover, during sampling at each stage, the two principles of short time and small effect were ensured as much as possible. The premise of sampling is that the soil exhibits sufficient consolidation strength to support the sampling without affecting the surrounding soil. Owing to the high moisture content and poor consolidation of the soil in the vacuum phase of SVP at 10 kPa, sampling and permeability testing are not conducted in the vacuum phase of the technique at 10 kPa. The soil at the bottom of the settler is also neither sampled nor evaluated to prevent disturbance, except at the end of the experiment. (Buddhima Indraratna, Chamari Bamunawita, Hadi Khabbaz. Numerical modeling of vacuum preloading and field applications. Canadian Geotechnical Journal, 2004, 41, 1098-1110) (Yuan, X.Q.; Wang, Q.; Lu, W.X.; Zhang, W.; Chen,H.E.; Zhang Y. Indoor simulation test of step vacuum preloading for high-clay content dredger fill. Marine Georesources and Geotechnology. 2018, 36, 83-90) (Fang, Y.G.; Guo, L.F.; Huang, J.W. Mechanism test on inhomogeneity of dredged fill during vacuum preloading consolidation. Marine Georesources and Geotechnology. 2019, 37, 1007-1017.)

Reviewer 2 Report
Experimental study on the effectiveness of the step vacuum preloading method in consolidating dredger fill with high clay content is presented in the paper.
But to use just two samples for experimentally study is not enough. The clay sample with similar properties was used in the paper ref. n. 23:
Yuan, X.Q.; Wang, Q.; Lu, W.X.; Zhang, W.; Chen, H. Conclusions in this paper are the same except for permeability measurement.
The only new conclusion from the experimental study is that the permeability coefficient of the soil decreases gradually with consolidation time.
Comments:
line 17: The results indicate that the settlement and consolidation of the soil are great. Could you please explain what does it mean "geat!"?
line 20 and 21: Could you please explain what does it mean "the difference is not so much"?
Conclusions:
The settlement, moisture content, drainage rate, and pore water pressure of the soil during reinforcement were monitored and already evaluated for step vacuum preloading method in ref. 23. with the same conclusions. What is the added value of your experimental study?
Author Response
Thank you very much for your valuable comments. 1.line 17: The results indicate that the settlement and consolidation of the soil are great. Could you please explain what does it mean "geat!"? Response: The sentence "The results indicate that the settlement and consolidation of the soil are great" in line 17 has been changed to "The results indicate that soil settlement increases at each stage of vacuum pressure treatment, and the degree of vertical consolidation at each stage exceeds 90%." 2.line 20 and 21: Could you please explain what does it mean "the difference is not so much"? Response: The sentence "the difference is not so much "has been changed to "but the permeability coefficients at two locations are always within the same order of magnitude." 3.But to use just two samples for experimentally study is not enough. Response: Taking more samples can establish the change trend of soil parameters with different distances and heights. However, selecting an excessive number of samples will easily affect the surrounding soil mass, resulting in inaccuracies. 4.The settlement, moisture content, drainage rate, and pore water pressure of the soil during reinforcement were monitored and already evaluated for step vacuum preloading method in ref. 23. with the same conclusions. What is the added value of your experimental study? Response: During testing, the vacuum loading mode (10, 20, 40, 60, and 80 kPa) is optimized, and changes in various parameters during vacuum-pressure treatment are carefully studied. This study provides a comprehensive analysis of the effectiveness of SVP. On the basis of the changes in drainage mode (free water or loosely bound water), horizontal displacement, and permeability coefficient, this study comprehensively analyzes the change in the internal properties of soil under SVP, which is not available in the study by Yuan.
